# Potentiation of β-Lactams against Methicillin-Resistant *Staphylococcus aureus* (MRSA) Using Octyl Gallate, a Food-Grade Antioxidant

**DOI:** 10.3390/antibiotics11020266

**Published:** 2022-02-18

**Authors:** Migma Dorji Tamang, Junghee Bae, Myungseo Park, Byeonghwa Jeon

**Affiliations:** 1School of Public Health, University of Alberta, Edmonton, AB T6G 2R3, Canada; migmatamang@gmail.com (M.D.T.); biblehee@hotmail.com (J.B.); 2Division of Environmental Health Sciences, School of Public Health, University of Minnesota, St. Paul, MN 55108, USA; park2421@umn.edu

**Keywords:** methicillin-resistant *Staphylococcus aureus*, drug potentiation, antioxidants

## Abstract

Methicillin-resistant *Staphylococcus aureus* (MRSA) is resistant to a number of antibiotics of clinical importance and is a serious threat to public health. Since bacteria rapidly develop resistance even to newly discovered antibiotics, this study aimed to develop drug potentiators to enhance the antibacterial activity of existing antibiotics for the control of MRSA. Based on our previous studies, screening of antimicrobial synergy was conducted with gallic acid and its derivatives using checkerboard assays. Antimicrobial synergy was confirmed with MRSA isolates from clinical cases. Combinations of penicillin, ampicillin, and cephalothin with octyl gallate (OG), an antioxidant approved by the US Food and Drug Administration (FDA), consistently exhibited synergistic bacteriostatic and bactericidal activities against MRSA, rendering MRSA sensitive to β-lactams. The fractional inhibitory concentration (FIC) and fractional bactericidal concentration (FBC) indices exhibited that the antimicrobial effects of OG were synergistic. The results of a permeability assay showed that OG significantly increased the permeability of the bacterial cell wall. Despite the intrinsic resistance of MRSA to β-lactams, the findings in this study demonstrated that OG enhanced the activity of β-lactams in MRSA and sensitized MRSA to β-lactams, suggesting that OG can be used as a drug potentiator to control MRSA using existing antibiotics.

## 1. Introduction

Multidrug-resistant bacterial infections are a rapidly growing threat to public health worldwide [1,2]. Particularly, methicillin-resistant *Staphylococcus aureus* (MRSA) is resistant to many conventional antibiotic therapies and frequently implicated in fatal infections, such as bacteremia, endocarditis, and pneumonia, and poses a significant threat to public health [3]. MRSA is resistant to all β-lactams and other clinically important antibiotics, such as linezolid and daptomycin, and increasingly resistant to the last-resort antibiotics, such as vancomycin [4,5,6,7], further limiting therapeutic options to treat MRSA infections. Accounting for more than 80,000 hospital-acquired infections in the United States, MRSA rapidly spreads from healthcare facilities and results in the emergence of community-associated pathogens (i.e., CA-MRSA) [8]. Although several new antimicrobial agents have been introduced to control MRSA infections, MRSA remains difficult to treat [9].

New antibiotics are urgently needed for the treatment of antibiotic-resistant infections. However, bacteria rapidly develop resistance even to newly-discovered antibiotics [10]. Alternatively, drug potentiation is considered as a novel approach to the control of antibiotic resistance. Drug potentiators, also called antibiotic adjuvants, are not necessarily antimicrobials but enhance the susceptibility of antibiotic-resistant pathogens to antibiotics [11]. For instance, clavulanic acid is not antimicrobial but is used with amoxicillin, a β-lactam drug, because clavulanic acid inhibits the function of β-lactamases [12]. Previously, we demonstrated that the antimicrobial and anti-biofilm activities of bacitracin, an antimicrobial peptide, were synergistically increased in MRSA when combined with octyl gallate (OG), a food additive approved by the U.S Food and Drug Administration (FDA) to prevent lipid oxidation [13,14]. In the present study, we determined antimicrobial synergy between OG and other antibiotic classes and discovered that OG significantly potentiated β-lactams against MRSA.

## 2. Results

### 2.1. Synergy between OG and Antibiotics against MRSA and Methicillin-Sensitive Staphylococcus aureus (MSSA)

Since our previous studies showed that some gallic acid derivates generated antimicrobial synergy with bacitracin [13,14], in this study, we investigated whether gallic acid derivatives may increase antimicrobial activity in MRSA when used in combination with antibiotics of other classes. The initial screening was conducted by examining the antimicrobial activities of erythromycin, ampicillin, gentamicin, kanamycin, and ciprofloxacin in combination with gallic acid, methyl gallate, ethyl gallate, propyl gallate, butyl gallate, OG, dodecyl gallate, and stearyl gallate. Although OG and dodecyl gallate exhibited antimicrobial synergy, OG produced significant antimicrobial synergy in repeated experiments (data not shown); thus, we focused on OG in the rest of the study.

When combined with antibiotics, OG synergistically increased their bacteriostatic and bactericidal activities in MRSA ATCC 33593 (Table 1). Particularly, MRSA ATCC 33593 was markedly sensitized to β-lactams, such as penicillin, ampicillin, and cephalothin (Table 1). The presence of 4 μg/mL of OG reduced the minimum inhibitory concentration (MIC) of ampicillin by 128-fold and the minimum bactericidal concentration (MBC) by 64-fold (Table 1). OG also generated antimicrobial synergy in methicillin-sensitive *Staphylococcus aureus* (MSSA) to a less degree than the synergy observed in MRSA based on the fold changes in MICs (Table 1). The results demonstrated that OG potentiates antibiotics, particularly β-lactams, against both MRSA and MSSA.

### 2.2. Antimicrobial Synergy of OG with β-Lactams in Clinical Isolates of MRSA

The synergistic antimicrobial activity of OG and β-lactam combinations was examined with clinical isolates of MRSA. The MICs of penicillin and ampicillin in clinical isolates of MRSA dramatically decreased by 128–1024-folds in the presence of 4 μg/mL OG (Table 2). Based on the fractional inhibitory concentration (FIC) index, the increased antimicrobial activity in β-lactams by OG was synergistic (Table 2) as an FIC index < 0.50 is deemed as synergy [15]. Synergistic bactericidal activity was determined by examining the viability of MRSA isolates on MH agar plates after exposure to OG and β-lactams (Figure 1). The degree of antimicrobial synergy was substantial in all the tested strains, even when a more stringent fractional bactericidal concentration (FBC) index for synergy (≤0.25) was applied (Table 3). The results showed that OG restored the antibacterial activity of penicillin, ampicillin, and cephalothin against MRSA.

### 2.3. OG Increased Cell Wall Permeability in MRSA

The findings above demonstrated that OG generated significant antimicrobial synergy with β-lactams. Previously, we reported that OG substantially increased the antimicrobial activity of bacitracin in MRSA [13]. Since the modes of action of both β-lactams and bacitracin are commonly related to the inhibition of peptidoglycan synthesis, we hypothesized that antimicrobial synergy with OG may involve alterations in permeability in the bacterial cell wall. To test the hypothesis, a permeability assay was conducted with propidium iodide (PI), a fluorescent dye [16]. PI can permeate the compromised bacterial membrane and binds to DNA [17]. Since the fluorescence of PI substantially increases after intercalating into DNA, the degree of a fluorescence signal represents membrane permeability in bacteria. Consistent with our hypothesis, cell wall permeability in MRSA was instantly increased by the addition of even low concentrations of OG (Figure 2), indicating that OG increases the membrane permeability in MRSA. The results suggest that OG may potentiate β-lactams in MRSA by permeabilizing the bacterial cell wall.

## 3. Discussion

The discovery of antibiotic potentiators is a novel strategy to extend the utility of existing antibiotics [18,19]. A number of studies have reported that certain non-antimicrobial compounds can intensify antimicrobial activity against MRSA when combined together. For instance, repurposing bithionol, an anthelmintic drug, has been suggested to control MRSA because bithionol kills MRSA persister cells by permeabilizing lipid bilayers and potentiates gentamicin against MRSA [20]. In a study conducted by Yarlagadda et al., venturicidin A, a non-antibiotic compound produced by actinomycetes isolated from soil, was shown to potentiate aminoglycosides against MRSA by interrupting ATP synthesis [21]. These studies commonly discovered potentiators enhancing the activity of gentamicin drugs in MRSA, and it is rare to discover potentiators that synergistically increase the activity of β-lactams in MRSA other than β-lactamase inhibitors, such as clavulanic acid [12].

In this study, we discovered that OG significantly enhanced the activities of β-lactams, such as benzylpenicillin, aminopenicillin, and cephalosporin drugs. OG is not a β-lactamase inhibitor but is a food additive approved by the US FDA to prevent the oxidation of high-fat foods, such as margarine [22]. Although the mode of action of OG in antimicrobial synergy has not yet been fully elucidated, our findings suggest that OG increases the permeability of the bacterial cell wall in MRSA (Figure 2). Previously, we reported that OG and dodecyl gallate have relatively lower MICs (32 μg/mL for both) in MRSA, whereas the MICs of gallic acid and its other derivatives are >128 μg/mL [13]. Perhaps, the strong permeabilizing activity of OG, as seen in Figure 2, may make this compound antimicrobial when used at high concentrations. Shi et al. [23] developed a delivery system by constructing an inclusion complex between OG and β-cyclodextrin, a cyclic oligosaccharide with a hydrophilic outer surface and a hydrophobic cavity, and increased the solubility and antimicrobial activity of OG against *Pseudomonas fluorescens* and *Vibrio parahaemolyticus*. Here, our data suggest that OG can be used as a potentiator of β-lactams to control MRSA when used at low concentrations. Stapleton et al. [24] reported that catechins and gallates present in aqueous extracts of Japanese green tea can reversed oxacillin resistance in MRSA but OG did not modulate oxacillin resistance. Since our study did not include oxacillin, we cannot explain why OG did not generate synergy with oxacillin. Possibly it can be ascribed to the use of a different antibiotic in different experimental settings.

According to our present and previous studies, OG commonly potentiates antimicrobials targeting the bacterial cell wall. β-Lactams inhibit the synthesis of the peptidoglycan layer, and bacitracin is an antimicrobial peptide interrupting peptidoglycan biosynthesis by targeting undecaprenyl pyrophosphate (UPP) [25]. Presumably, OG permeabilizes the bacterial cell wall, which facilitates the access of bacitracin and β-lactams to their cellular targets and generates antimicrobial synergy. Future studies are still required to explain molecular mechanisms for the synergy. Nevertheless, the findings in this study suggest that OG is a promising drug potentiator capable of re-sensitizing MRSA to β-lactams, which are not currently used for the treatment of MRSA infection due to the resistance of MRSA to this antibiotic class.

## 4. Materials and Methods

### 4.1. Bacterial Strains and Culture

MRSA ATCC 33593 and methicillin-sensitive *S. aureus* (MSSA) ATCC 29213 were purchased from the American Type Culture Collection (ATCC; Manassas, VA, USA). Human clinical strains of MRSA 3795, MRSA 3823, MRSA 3865, and MRSA 3903 were purchased from the Culture Collection of Antimicrobial Resistant Microbes at the Korea National Research Resource Center (Seoul, Korea). All bacterial strains were routinely cultured on Trypticase Soy (TS) medium at 37 °C.

### 4.2. Antimicrobial Susceptibility Testing

MIC was determined with a broth microdilution method by growing MRSA on Mueller–Hinton (MH) broth (BBL, Sparks, MD, USA) according to Clinical and Laboratory Standard Institute (CLSI) guidelines [26]. *S. aureus* ATCC 29213 was used as a quality control strain. Antibiotics (penicillin, ampicillin, cephalothin, gentamicin, chloramphenicol, tetracycline, florfenicol, erythromycin, and lincomycin) and OG were purchased from MilliporeSigma (St Louis, MO, USA). The test can measure antimicrobial activity in planktonic cells. MBC was determined by spotting 10 μL of culture from a 96-well plate used to measure MICs onto MH agar plates.

### 4.3. Checkerboard Titration Assay

The synergy of OG with antibiotics was determined with checkerboard titration assays as described previously [27]. Briefly, antibiotics and OG were two-fold serially diluted on column and row, respectively. Each well was inoculated with 100 µL of MRSA suspension (~5 × 10^4^ CFU per well), and checkerboard plates were incubated aerobically at 37 °C for 24 h. Checkerboard assays for each combination were performed at least three times.

### 4.4. Calculation of FIC and FBC Indices

Synergistic effects were evaluated by determining an FIC index as described [28]. The FIC index was calculated as FIC = FIC_A_ + FIC_B_, where FIC_A_ is the MIC of agents A and B in combination divided by the MIC of agent A alone, and FIC_B_ is the MIC of agents A and B in combination divided by the MIC of Agent B alone. An FIC index of ≤0.5 indicates synergistic activity [15]. An FBC index was similarly determined based on the MBCs of OG and antibiotic combinations.

### 4.5. Cell Wall Permeability Assay

The effect of OG on cell wall permeability was examined as described previously with minor modifications [16]. Briefly, an overnight culture of *S. aureus* ATCC33593 was washed with 5 mM HEPES buffer containing 20 mM glucose, pH 7.2. The culture was diluted with the same buffer to OD_600_ = 0.5 and incubated with propidium iodide (7.5 μg/mL). After 10 min, OG was added to the bacterial suspension at different concentrations, and fluorescence was measured with a plate reader (Varioskan, ThermoFisher, Waltham, MA, USA) with 535 nm excitation and 615 nm emission. The level of permeability was adjusted by subtracting the fluorescence intensity of a negative control measured without bacterial cells. The experiment was repeated three times.

## Figures and Tables

**Figure 1 antibiotics-11-00266-f001:**
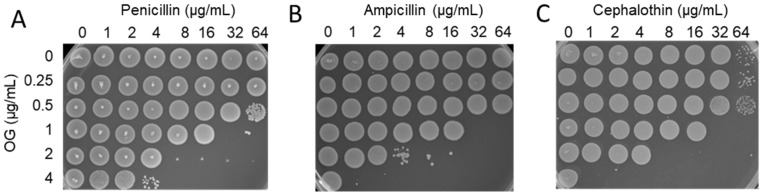
Synergistic bactericidal activity of octyl gallate (OG) in combination with β-lactams, including (**A**) penicillin, (**B**) ampicillin, and (**C**) cephalothin.

**Figure 2 antibiotics-11-00266-f002:**
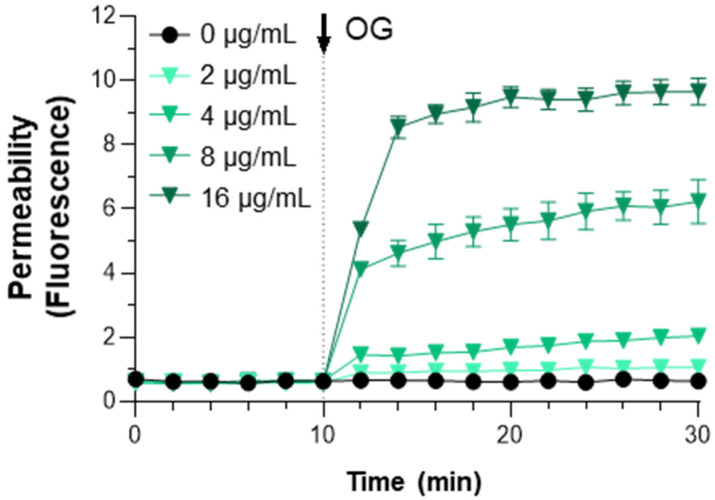
Alterations in cell wall permeability in MRSA by octyl gallate (OG). The permeability was measured with propidium iodide. The arrow indicates the time point when OG was added to bacterial suspensions. The results show the means and standard deviations of fluorescence values from three samples in a single experiment. The experiment was repeated three times and produced similar results.

**Table 1 antibiotics-11-00266-t001:** Synergistic bacteriostatic and bactericidal activities of octyl gallate (OG) and antibiotic combinations.

Antibiotic	MIC (µg/mL) ^ⱡ^			MBC (µg/mL) ^ⱡ^		
	Antibiotic Alone	Antibiotic Plus OG	FIC Index	Antibiotic Alone	Antibiotic Plus OG	FBC Index
**MSSA**						
Penicillin	1	0.25 (4)	0.094	1	0.25 (4)	0.281
Ampicillin	2	0.125 (16)	0.125	2	0.125 (16)	0.094
Cephalothin	0.5	0.125 (4)	0.5	0.5	0.125 (4)	0.258
Gentamicin	1	0.125 (8)	0.129	1	0.125 (8)	0.144
Chloramphenicol	16	2 (8)	0.329	32	2 (16)	0.082
Tetracycline	1	0.125 (8)	0.129	2	0.063 (32)	0.051
Erythromycin	0.5	0.25 (2)	0.539	2	0.25 (8)	0.144
Lincomycin	1	0.5 (2)	0.539	2	0.25 (8)	0.144
**MRSA**						
Penicillin	>64	0.25 (>256)	0.127	>64	8 (>8)	0.126
Ampicillin	>64	0.5 (>128)	0.129	>64	1 (>64)	0.066
Cephalothin	>64	1 (>64)	0.133	64	2 (32)	0.070
Gentamicin	>64	8 (>8)	0.156	>64	16 (>4)	0.047
Chloramphenicol	16	1 (16)	0.188	32	2 (16)	0.125
Tetracycline	>64	16 (>4)	0.188	>64	64 (>1)	0.375
Erythromycin	>64	>4 (>16)	>2	>64	>4 (>16)	>2
Lincomycin	>64	>16 (>4)	>2	>64	>16 (>4)	>2

ⱡ: Fold changes in the MIC and MIC of antibiotics and 4 μg/mL OG combinations compared with those of antibiotics alone are indicated in parentheses.

**Table 2 antibiotics-11-00266-t002:** MIC and FIC index of octyl gallate (OG) and β-lactam combinations against MRSA isolates from clinical cases in humans.

Strain	MIC (μg/mL)			MIC of Combinations (μg/mL) ^ⱡ^			FIC Index		
	PEN	AMP	CEP	PEN + OG	AMP + OG	CEP + OG	PEN + OG	AMP + OG	CEP + OG
MRSA 3795	>64	>64	>64	0.063 (>1024)	0.125 (>512)	1 (>64)	0.032	0.032	0.039
MRSA 3823	>64	>64	>64	0.063 (>1024)	0.125 (>512)	1 (>64)	0.126	0.126	0.133
MRSA 3865	64	>64	>64	0.5 (128)	1 (>64)	4 (>16)	0.070	0.070	0.156
MRSA 3903	>64	>64	>64	0.125 (>512)	0.125 (>512)	1 (>64)	0.126	0.126	0.133

PEN: penicillin, AMP: ampicillin, CEP: cephalothin, OG: octyl gallate. ⱡ: Fold changes in the MICs and MICs of antibiotics and 4 μg/mL OG combinations in comparison with those of antibiotics alone are indicated in parentheses.

**Table 3 antibiotics-11-00266-t003:** MBC and FBC index of octyl gallate (OG) and β-lactam combinations against MRSA isolates from clinical cases in humans.

Strain	MBC (μg/mL)			MBC of Combinations (μg/mL) ^ⱡ^			FBC Index		
	PEN	AMP	CEP	PEN + OG	AMP + OG	CEP + OG	PEN + OG	AMP + OG	CEP + OG
MRSA 3795	>64	>64	>64	0.063 (>1024)	0.25 (>256)	1 (>64)	0.063	0.063	0.035
MRSA 3823	>64	>64	>64	0.5 (>128)	0.5 (>128)	2 (>32)	0.033	0.127	0.039
MRSA 3865	64	>64	>64	4 (16)	8 (>8)	8 (>8)	0.188	0.156	0.156
MRSA 3903	>64	>64	>64	0.5 (>128)	0.5 (>128)	2 (>32)	0.033	0.033	0.039

PEN: penicillin, AMP: ampicillin, CEP: cephalothin, OG: octyl gallate. ⱡ: Fold changes in the MICs and MICs of antibiotics and 4 μg/mL OG combinations in comparison with those of antibiotics alone are indicated in parentheses.

## Data Availability

The data presented in this study are available on request from the corresponding author.

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
