# Peer review of "Potentiation of β-Lactams against Methicillin-Resistant Staphylococcus aureus (MRSA) Using Octyl Gallate, a Food-Grade Antioxidant"

_antibiotics, 2022, doi:10.3390/antibiotics11020266_

Round 1
Reviewer 1 Report
The manuscript is interesting, but the tests used are superficial. There is no explanation of the OG's mechanism of action. Authors suggest a synergistic effect of OG with antibiotics but cannot prove this action. What is the hypothesis of how the OG would be changing the wall's permeability?
It is unclear from the text whether planktonic or biofilm cultures were used. The permeability test is unclear, and information is lacking. How was permeability observed? If iodide is to quantify internalization, why is it used before? The conclusion was missing.
Author Response
- The manuscript is interesting, but the tests used are superficial. There is no explanation of the OG's mechanism of action. Authors suggest a synergistic effect of OG with antibiotics but cannot prove this action. What is the hypothesis of how the OG would be changing the wall's permeability?
Response: Thank you for the comment. The focus of the study was to identify antibiotics that generate synergy with gallic acid derivatives based on our previous reports as explained in our manuscript. To identify synergistic combinations with OG, a significant amount of testing was conducted by cross-combining eight different antibiotics with eight compounds, although the presented data are mainly about OG and beta-lactam combinations. Please understand that the purpose of the presented study is to screen and identify synergistic combinations of antibiotics and antioxidants. Currently, we are trying to understand the molecular mechanisms underlying synergy using RNA-seq. It is a whole different set of works. Considering the reviewer’s comment, we changed a discussion part (lines 171-176)
- It is unclear from the text whether planktonic or biofilm cultures were used.
Response: MIC and checkerboard assays can measure antimicrobial activity in planktonic cells. To evaluate anti-biofilm activity, biofilms grown on a plate surface should be stained with a dye, such as crystal violet, for visualization. We made it clear in the revised version (lines 197, 198)
- The permeability test is unclear, and information is lacking. How was permeability observed? If iodide is to quantify internalization, why is it used before? The conclusion was missing.
Response: According to the reviewer’s comment, we add details about the experiment and conclusion to the revised version (lines 113-117, 119).
Thank you!
Reviewer 2 Report
- As seen from the figure 1, intrinsic OG antimicrobial activity, i.e. without antibiotic, was tested in checkerboard assay. However, I suppose for clearer understanding it should additionally be noted in the text.
- In description of results (Tab. 1 and 2) 4 μg/ml of OG is mentioned: according to Fig.1, it was the maximal concentration tested, FIC with smaller concentration didn’t indicate synergy? Authors didn’t try higher OG concentration? Looks like it could work even better. As was shown in figure 2, bacterial cell permeability was significantly increased started from OG concentration of 4 mg/L.
- Is there any information about OG human pharmacokinetics? It is important to understanding how OG concentration of 4 mg/L related to the antioxidant concentrations in serum. Is it achievable?
- Line 55: I suppose the “ATCC MRSA and MSSA” should be specified in the “2.1. Synergy between OG and antibiotics against MRSA” section title.
Small edits:
Line 35: ref. 4 and 5 are related with other antibiotics, not vancomycin, maybe authors should place these references earlier?
Line 37: … and rapidly spread from healthcare facilities… - MRSA spreads?
Line 43: …drug potentiation is considered a novel approach to the control... - is considered as a? novel approach
Line 45: enhance the sensitivity – I would say “susceptibility” is more common
Author Response
- As seen from the figure 1, intrinsic OG antimicrobial activity, i.e. without antibiotic, was tested in checkerboard assay. However, I suppose for clearer understanding it should additionally be noted in the text.
Response: Considering the reviewer’s comment, we explained some intrinsic features of OG related to antimicrobial synergy in the revised version (lines 156-170). Thank you for the comment.
- In description of results (Tab. 1 and 2) 4 μg/ml of OG is mentioned: according to Fig.1, it was the maximal concentration tested, FIC with smaller concentration didn’t indicate synergy? Authors didn’t try higher OG concentration? Looks like it could work even better. As was shown in figure 2, bacterial cell permeability was significantly increased started from OG concentration of 4 mg/L.
Response: We agree that synergy could have been improved if we had used a higher concentration of OG. Since OG already generated significant synergy when used at low concentrations, such as 4 ug/ml, as seen in Tables 1, 2, and 3, and Figure 1, we did not increase the concentration. As replied to the comment above, OG is antimicrobial when used at high concentrations. The purpose of the study was to develop synergistic combinations by using low concentrations of antibiotics and antioxidants.
- Is there any information about OG human pharmacokinetics? It is important to understanding how OG concentration of 4 mg/L related to the antioxidant concentrations in serum. Is it achievable?
Response: Thank you for the important question. Answering your comment is a prerequisite to developing therapeutic applications of OG in humans. Currently, we have not found any information about OG human pharmacokinetics.
- Line 55: I suppose the “ATCC MRSA and MSSA” should be specified in the “2.1. Synergy between OG and antibiotics against MRSA” section title.
Response: Thank you for the comment. We correct the section title accordingly (lines 55, 56).
Small edits:
- Line 35: ref. 4 and 5 are related with other antibiotics, not vancomycin, maybe authors should place these references earlier?
Response: We added more references. Thank you (line 35, References 6 and 7).
- Line 37: … and rapidly spread from healthcare facilities… - MRSA spreads?
Response: We changed the sentence (lines 36-38)
- Line 43: …drug potentiation is considered a novel approach to the control... - is considered as a? novel approach
Response: We changed it (line 43).
- Line 45: enhance the sensitivity – I would say “susceptibility” is more common
Response: We changed it in the revised version (line 45). Thank you!
Reviewer 3 Report
The authors of the manuscript " Potentiation of β-lactams against methicillin-resistant Staphylococcus aureus (MRSA) using octyl gallate, a food-grade antioxidant" present their study clearly. The manuscript also contains appropriate references.
Author Response
- The authors of the manuscript " Potentiation of β-lactams against methicillin-resistant Staphylococcus aureus (MRSA) using octyl gallate, a food-grade antioxidant" present their study clearly. The manuscript also contains appropriate references.
Response: I appreciate your positive comments. Thank you so much!
Reviewer 4 Report
The authors cited the potentiation of β-lactams using octyl gallate against methicillin-resistant Staphylococcus aureus (MRSA). The authors performed systematic experiments, which are essential to prove their hypothesis. However, there are certain points that can be improved.
(i) There are some typographical mistakes as in Table 2 and Table 3, the top annotations are MIC μg/ml ) and MBC μg/ml ), which should be (MIC μg/ml ) and (MBC μg/ml).
(ii) Authors said on page 2 line 44, about the definition of antibiotic potentiators. Through the experiments, authors found octyl gallate increases the potentiations of Beta-lactam antibiotics. However, there are studies that say the Gallates or similar compounds are themselves sufficiently act as antimicrobial (e.g International Journal of Antimicrobial Agents Volume 23, Issue 5, May 2004, Pages 462-467 ) or possess antibacterial activity ( International Journal of Food Microbiology Volume 361, 16 January 2022, 109460). Please do provide clarification or add a small paragraph to clarify this point.
The manuscript illustrates sufficient material study and methodology. The authors considered a range of bacterial strains and, furthermore investigates the mode of the synergy of Beta-lactam with octyl gallate, therefore this manuscript would appeal to the researchers working in various fields of microbiology and, certainly falls in the interest of the current Journal.
Author Response
The authors cited the potentiation of β-lactams using octyl gallate against methicillin-resistant Staphylococcus aureus (MRSA). The authors performed systematic experiments, which are essential to prove their hypothesis. However, there are certain points that can be improved.
- (i) There are some typographical mistakes as in Table 2 and Table 3, the top annotations are MIC μg/ml ) and MBC μg/ml ), which should be (MIC μg/ml ) and (MBC μg/ml).
Response: Thank you. We corrected them in the revised version (Tables 2 and 3).
- (ii) Authors said on page 2 line 44, about the definition of antibiotic potentiators. Through the experiments, authors found octyl gallate increases the potentiations of Beta-lactam antibiotics. However, there are studies that say the Gallates or similar compounds are themselves sufficiently act as antimicrobial (e.g International Journal of Antimicrobial Agents Volume 23, Issue 5, May 2004, Pages 462-467 ) or possess antibacterial activity ( International Journal of Food Microbiology Volume 361, 16 January 2022, 109460). Please do provide clarification or add a small paragraph to clarify this point.
Response: Thank you for suggesting the articles. In the revised version, we provided clarification by comparing with the studies (lines 160-170). Thank you!
- The manuscript illustrates sufficient material study and methodology. The authors considered a range of bacterial strains and, furthermore investigates the mode of the synergy of Beta-lactam with octyl gallate, therefore this manuscript would appeal to the researchers working in various fields of microbiology and, certainly falls in the interest of the current Journal.
Response: Thank you!
Round 2
Reviewer 1 Report
Thanks for the clarifications from the authors.
All requests have been fulfilled.